# Deep Learning-Based Bearing Fault Diagnosis Method for Embedded Systems

**DOI:** 10.3390/s20236886

**Published:** 2020-12-02

**Authors:** Minh Tuan Pham, Jong-Myon Kim, Cheol Hong Kim

**Affiliations:** 1School of Electronics and Computer Engineering, Chonnam National University, Gwangju 61186, Korea; 196325@jnu.ac.kr; 2School of IT Convergence, University of Ulsan, Ulsan 44610, Korea; jmkim07@ulsan.ac.kr; 3School of Computer Science and Engineering, Soongsil University, Seoul 06978, Korea

**Keywords:** fault diagnosis, bearing fault, machine health monitoring, acoustic emission signals, signal decomposition, convolutional neural network, embedded systems

## Abstract

Bearing elements are vital in induction motors; therefore, early fault detection of rolling-element bearings is essential in machine health monitoring. With the advantage of fault feature representation techniques of time–frequency domain for nonstationary signals and the advent of convolutional neural networks (CNNs), bearing fault diagnosis has achieved high accuracy, even at variable rotational speeds. However, the required computation and memory resources of CNN-based fault diagnosis methods render it difficult to be compatible with embedded systems, which are essential in real industrial platforms because of their portability and low costs. This paper proposes a novel approach for establishing a CNN-based process for bearing fault diagnosis on embedded devices using acoustic emission signals, which reduces the computation costs significantly in classifying the bearing faults. A light state-of-the-art CNN model, MobileNet-v2, is established via pruning to optimize the required system resources. The input image size, which significantly affects the consumption of system resources, is decreased by our proposed signal representation method based on the constant-Q nonstationary Gabor transform and signal decomposition adopting ensemble empirical mode decomposition with a CNN-based method for selecting intrinsic mode functions. According to our experimental results, our proposed method can provide the accuracy for bearing faults classification by up to 99.58% with less computation overhead compared to previous deep learning-based fault diagnosis methods.

## 1. Introduction

Bearings are vital components in rotating machinery as they reduce the friction coefficient of the moving process. Due to harsh working environments, bearings are frequently deteriorated and make up more than 50% of machinery faults [1,2]. Unfortunately, serious bearing faults cause the rotating machines to waste maintenance time and costs and could trigger a chain of dangerous reactions [3,4,5]. Manual testing for the bearings makes it difficult to determine the best time for the maintenance. By contrast, real-time monitoring enables timely actions for bearing faults, resulting in improved reliability of the rotating machines.

The data-driven approach using acoustic emission (AE) signals has been proven to provide the advantage of capturing bearing health information from low-energy signals in realistic working conditions such as early developed damages, slow rotating machinery, and severe vibration environments [6,7,8,9,10]. AE-based methods are expected to replace the conventional methods using vibration signals [3,11] or current signals of electrical motors [12,13,14] in the maintenance field. In this work, we introduce a novel data-driven AE-based diagnosis method for bearing faults.

Fault features in bearings are difficult to identify because of time and frequency shifting characteristics. Therefore, resource-conserving approaches based on traditional signal processing methods, such as wavelet packet transforms and envelope analysis processing, are difficult to provide high diagnosis accuracy [15,16]. In recent years, several diagnosis methods for bearing faults have been proposed to tackle this problem based on artificial intelligence, especially convolutional neural networks (CNN) [17]. CNNs can be practical, automatic feature extraction tools for obtaining meaningful information of bearing faults. Hasan et al. proposed a scheme for bearing fault diagnosis based on the images from AE signals and a transfer learning to share the knowledge among data domains for achieving a high diagnosis accuracy under various operating conditions [18]. The bearing diagnosis method proposed by Tra et al. for compound faults under variable speeds used generic CNN architecture based on Lenet-5 [19] and enhanced the CNN training process using the stochastic diagonal Levenberg–Marquardt algorithm [20,21]. These methods represent AE signals by two-dimensional images and use CNN for feature extraction.

In general, the CNN-based approach consumes significant computational resources and memory capacity compared with conventional signal processing methods. For example, to represent 10,000 sampled signal points by a spectrogram in the time–frequency domain to create an image, the standard short-time Fourier transform (window length = 256; hop size = 64; number of FFT points = 256) consumes 89,158 multiplier–accumulator units (MACs) in MATLAB. In comparison, state-of-the-art CNN architectures such as EfficientNet-B0 [22] consume 0.195B of MACs to obtain the prediction results after inferring the input image. Recent CNN-based diagnosis methods using extremely deep CNN architectures can achieve high reliability even in complex operating conditions but are only suited for the industrial PC system. Compared with typical industrial-PC-based solutions for controlling bearings hazards, embedded platforms offer advantages of small size, portability, low cost, and low power consumption, thereby accommodating the industrial operating environments. Nevertheless, due to limited resources, embedded systems cannot perform complex algorithms to achieve high accuracy for bearing fault diagnosis in complex operating conditions (e.g., variant motor speeds in a wind turbine system [23] and changing loads). Simple CNN architectures for bearing fault diagnosis require fewer system resources, but they cannot guarantee high reliability and stability in diagnosis, particularly when noisy conditions are involved. This motivated us to study a new CNN-based method for the classification of bearing faults on embedded systems.

To obtain an accurate prediction process while maintaining the low cost of system resources for embedded systems, we propose a new bearing fault diagnosis process. We propose a new approach leveraging the advantages of the state-of-the-art CNN model, MobileNet-v2, which utilizes modern CNN design techniques for mobile devices and embedded systems. To reduce the required computation resources in our proposed diagnosis method, a pruning technique is applied to remove unnecessary trained network components after the training process. Eventually, our proposed CNN model can provide a high diagnostic accuracy with little computation resources, which is suitable for embedded systems. Moreover, based on the observation that the input image size significantly affects the computation complexity of the CNNs, we also decompose raw AE signals to remove less critical components. Our method helps small-size images to represent the features of bearing faults from AE signals. Decomposition is processed by ensemble empirical mode decomposition (EEMD) and our proposed CNN-based method to select intrinsic mode functions (IMFs). Spectrogram images are created by using the constant-Q transform (CQT) to utilize the time–frequency domain representation ability for the nonstationary signals. Experimental results show that our proposed method provides high diagnostic accuracy for bearing fault signals in complex conditions of variable rotational speeds. The remainder of this paper is organized as follows: Section 2 describes the proposed bearing fault diagnosis method for embedded systems. Section 3 presents our experimental methods, and Section 4 discusses experimental results. Finally, Section 5 provides a conclusion.

## 2. Proposed Bearing Fault Diagnosis Method

Figure 1 shows an overview of the proposed bearing fault diagnosis method. The proposed fault diagnosis method for bearings comprises two primary phases: offline and online. The offline phase is related to the CNN model establishment, whereas the online phase uses the established CNN model to perform the bearing fault diagnosis on embedded systems. The proposed fault diagnosis method for embedded systems is based on two main principles: (1) reducing the input size of the CNN model and (2) pruning unnecessary components of a trained CNN model. To minimize the size of input images, we propose a new representation method for bearing status information. First, due to AE signals’ nonstationary characteristics, we use a time–frequency domain transform (CQT: constant-Q transform) to create spectrogram images of AE signals. Moreover, when small images are used to represent the signals, a prior process for eliminating redundant signal components is necessary to avoid the sparsity of useful information (the images are resized by bilinear interpolation before feeding the CNN model). Redundant signal components can be noises, not containing bearing status information. Therefore, representing only the most useful features can reduce the feature extraction workload of the CNN model, which requires a small number of parameters, thereby reducing the number of FLOPs and inference time. To enable pruning unnecessary components of a trained CNN model, instead of designing shallow CNN architectures with primary convolution and pooling layers, we use an up-to-date CNN architecture. The adopted practical CNN model can extract the features from spectrogram images effectively. After the training, with accumulated knowledge regarding the training data, the pruning is performed to remove unnecessary parts of the trained model to reduce the required system resources while ensuring a high prediction accuracy.

The offline phase in the proposed fault diagnosis method comprises two stages: selecting useful intrinsic mode function (IMF) orders and establishing the bearing fault diagnosis model for embedded systems. In the first stage, we perform ensemble empirical mode decomposition (EEMD) [24] to decompose the raw signals into components, which are a series of IMFs. A useful IMF for bearing fault diagnosis must satisfy two conditions: (1) the selected IMF should exhibit a high correlation with the original signals, and (2) the selected IMF must consist of the features related to bearing fault status. Therefore, we first use a correlation coefficient (CC) scores to obtain IMF orders satisfying the first condition. The IMFs that satisfy the first condition are known as related IMFs (RIMFs). Subsequently, our proposed method uses a high-performance CNN model (EfficientNet-B0) to determine the RIMFs consisting of bearing status features rapidly. Note that the CNNs used to select the RIMFs will no longer be required for the online phase on embedded systems. Therefore, they do not affect the amount of workload in the online phase. At the end of the first stage, the IMF orders that are useful for bearing fault diagnosis can be determined. These useful IMF orders will be added up to construct the new signals to be used in the second stage as well as the online phase, thereby establishing the bearing diagnosis model. All of the other raw signals are decomposed according to the determined IMF orders and then transformed into the spectrograms by applying the constant-Q nonstationary Gabor transform [25]. Next, the spectrogram images are applied to image normalization techniques and used as training data for a state-of-the-art CNN model designed for mobile devices and embedded systems, i.e., MobileNet-v2 [26]. In the MobileNet-v2 training step, gradient centralization is used as an optimizer to accelerate the training process and improve the final generalization performance. After the training process is completed, the model is compressed by the auto-compress pruner. The auto compress pruner approach in designing the CNN model for fault diagnosis can be a good solution for embedded systems by reducing the size of the CNN model and the number of FLOPs. It can also improve the accuracy of the CNN model by pruning redundant parts of weight values, which are the meaningless constraints on the CNN model. At the end of the offline phase, a light-trained CNN model is generated for bearing fault classification in embedded systems. In the online phase, each signal segment from the testing subset is selected and transformed in the same manner as in the second step of the offline phase. After the spectrogram images are created in the online phase, they are resized to be compatible with the required input size of the light-trained CNN model. Finally, the light-trained CNN model automatically infers the bearing status corresponding to the AE input signals, enabling the accurate diagnosis for the bearing faults in embedded systems.

### 2.1. Nonstationary Gabor Frames and CQT

Some empirical formulas illustrate the correlation between defect frequencies of acquired signals from the bearing failures and rotating speed [27]. This correlation implies that the defect frequencies change with time under the condition of variable rotational speeds. That leads to the nonstationary characteristics of the signals containing bearing fault information. The time–frequency signal processing appears like a natural extension of the time and frequency domain processing, using the representations of signals in a space that can display the information from nonstationary signals in a more informative manner [28].

Gabor analysis [29] is widely used in signal processing applications, particularly for nonstationary signals. Nonstationary Gabor frames enable a perfect reconstruction. In standard Gabor analysis, a window of fixed size tiles the time–frequency plane, where a Gabor frame is a collection of windowing functions of various sizes that are used to tile the time–frequency plane. Nonstationary Gabor frames are useful for analyzing the types of signals, i.e., fixed-sized time–frequency windows. Unlike the short-time Fourier transform, the windows used in the CQT [25] have adaptable bandwidths and sampling densities. In the frequency space, the windows are centered at logarithmically spaced center frequencies.

In the proposed method for bearing fault diagnosis, Constant-Q Transform (CQT) is applied to produce the spectrograms representing the status of bearings, as shown in Figure 2. In the CQT, the bandwidth and sampling density in the frequency domain are varied [30]. The windows are constructed and applied directly to the frequency domain. Different windows have different center frequencies and bandwidths, but the center-frequency-to-bandwidth ratio remains constant. Maintaining a constant ratio implies that (1) the resolution in time improves at higher frequencies, and (2) the resolution in frequency improves at lower frequencies. The time shifts for each window depending on the bandwidth, owing to the uncertainty principle.

The CQT depends on the following: (1) window functions $g_{k}$, which are real-valued, even functions. In the frequency domain, the Fourier transform of $g_{k}$ is defined in the interval $\left[-F_{s}/2,F_{s}/2\right]$; (2) the sampling rate $\omega_{s}$; (3) the number of bins per octave, b; (4) the minimum and maximum frequencies, $\omega_{\min}$ and, $\omega_{\max}$ respectively.

### 2.2. Extracting IMFs by EEMD

The empirical mode decomposition (EMD) [31,32] algorithm was first proposed by a group of researchers from NASA to address the method to decompose a specific signal into simpler components without a priori information regarding the linearity or stationarity of the signal. The simpler components are known as intrinsic mode functions (IMFs), which satisfy two conditions: (1) the number of extrema and the number of zero crossings must be equal or differ by one at most; and (2) the mean envelope determined by the lower and upper envelopes must be zero. The EMD is a continuous process that selects the number of IMFs sequentially from the original signal. The EMD is sensitive to noises, so it seems to be unstable. Ensemble empirical mode decomposition (EEMD) [24] algorithm is proposed as an improved version of the EMD. The EEMD can overcome the sensitivity of noise effectively by utilizing noise as an assisted tool for analysis. The general idea of this algorithm is that white noise has a uniform distribution providing EMD a relatively homogeneous reference scale distribution. Due to the scale variety of background-white noise, the signals at different scales are mapped appropriately automatically to guarantee the continuity of each IMF in the time domain. Averaging multiple IMFs can eliminate the effects of white noise because the mean of background white noise is zero. In the EEMD, the original signal is added to sufficient groups of white noise realizations that have the same mean and variance values as the original signal. Moreover, then, the algorithm performs multiple decompositions. Each original signal added to a specific group of noise is decomposed by the EMD to obtain various groups of IMFs. The final decomposition is then computed as the average of all these decompositions [32].

Concerning the implementation of EEMD, two parameters are considered: (1) Nstd, which is the ratio between the standard deviation value of the added white noise and the standard deviation value of the original signal, (2) NE, which is the number of groups of noise to be added to the original signal. In this study, the first parameter Nstd was set to 0.2, according to the suggestion in [33], which demonstrated that the signals dominating at high frequencies might be affected by smaller noise amplitudes, and vice versa. In addition, after conducting some experiments, it was discovered that NE (the number of noise groups) set to 200 was sufficient.

### 2.3. Choosing Useful IMFs

As mentioned above, our approach for obtaining useful IMFs from a series of IMFs decomposed by applying the EEMD comprises two steps. The first step is calculating the correlation coefficients between the assigned IMFs and the original signals by using Equation (1)
(1)$$\lambda =\frac{\sum_{n=1}^{N}\left[x(t)c(t)\right]}{\sqrt{\sum_{n=1}^{N}x{\left(t\right)}^{2}\sum_{n=1}^{N}c{\left(t\right)}^{2}}}$$
where x(t) is the original signal, c(t) is an IMF, and N is the total number of sampling points. Top IMFs having high correlation coefficient (CC) scores compared with the remaining ones are selected into the RIMF group. Moreover, then, we create the datasets whose quantity is equal to the number of RIMFs. Each dataset comprises the signals extracted in a specific RIMF order. These AE signals are acquired from the bearings with single faults (ORD, IRD, RD). Thus, each dataset delegates only one RIMF, and useful IMFs from RIMFs can be selected by evaluating the features in the datasets. A RIMF is selected if its corresponding dataset shows a good performance in the task of classifying single bearing faults (the signals in these datasets are represented as the spectrograms by CQT before being evaluated by a CNN). In this study, a high-performance CNN model, Efficient-Net B0 [22], is used to evaluate the feature representation ability of those datasets.

### 2.4. Establishing CNN Model for Embedded Systems Using MobileNet-v2 and Pruning

MobileNet-v2 is considered an efficient CNN model for mobile devices (or embedded systems) with potential characteristics of small size, low latency, and low power [26]. Its architecture is optimized to achieve high accuracy while minimizing the number of parameters and computational resources, which is suitable for mobile devices. It is constructed based on the concept of MobileNet-v1, which is assembled from depth-wise separable convolution blocks (a type of factorized convolution that reduces computational costs compared with standard convolutions). The upgraded version, MobileNet-v2, comprises some new features: (1) linear bottlenecks between layers and (2) shortcut connections between bottlenecks. The bottlenecks encode the information between inputs and outputs; lower-level concepts such as pixels encapsulates effectively to higher-level descriptors such as image class. Residual connections act as shortcuts to enable a faster training process and get better performance.

The architecture is constructed from basic building blocks, which is a bottleneck depth-separable convolution with residuals. The architecture of MobileNet-v2 comprises an initial fully convolutional layer with 32 filters, followed by 19 residual bottleneck layers. This study tailored the architecture to different performance points by using variable input image resolutions and scaling rates (width multiplier), which are tunable hyperparameters and compromise the desired accuracy. The role of the scaling rate α is to thin a network uniformly at each layer. For a specified layer and scaling rate α, the number of input channels M becomes αM, and the number of output channels N becomes αN. After this, αM and αN are rounded to ensure that all layers have a divisible channel number that is divisible by 8 (the smallest number of channels is 8).

The disadvantage of state-of-the-art CNN architecture is that the model size increases significantly, which deteriorates memory storage and computational resources. Pruning is a practical model compression technique applied to the CNNs for reducing storage and computational resources while accelerating the model inference time [34]. However, previous pruning techniques require an amount of manual labor to determine the parts of trained weights and other hyperparameters that are redundant. Therefore, automatic determination processes have been proposed to effectively identify meaningless parts without high costs. Auto-compression is a practical algorithm that can be applied to recent popular backbone CNN architectures [35]. It can prune a model at a high rate while maintaining the same prediction accuracy. Moreover, its significant inference speedup renders it compatible with embedded-system-based deep learning applications in terms of real-time performance ability.

The generic flow of an automatic process is (1) action sampling, (2) quick action evaluation, (3) decision making, and (4) actual pruning and result generation. The auto-compression technique [35] is based on this flow and improves some stages to achieve the target of a maximum decrease in the number of FLOPs and a minimum reduction of accuracy. The improvement of the auto-compression algorithm is attributed to three primary reasons. The first is the combination of filter pruning and column pruning instead of only using filter pruning. The second is the use of a core algorithm related to the purification step of the alternating direction method of multipliers (ADMM)-based weight pruning algorithm. The third is the enhancement of a classical heuristic search technique known as simulated annealing instead of a deep reinforcement learning (DRL)-based framework to affect step (1) (action sampling) and step (3) (decision making) in the genetic flow.

With an initial trained CNN for fault classification, the auto-compression framework for structured weight pruning can be performed in two phases. Phase I is a structured prune based on ADMM, and Phase II is the purification. Each phase contains a number of rounds, and each weight pruning result of the previous round is used as an input for the next round. The two phases of the process are similar. The algorithm for each of these phases is illustrated by the flowchart in Figure 3 [35]. In the algorithm, the input is the trained CNN model based on MobileNet-v2 for bearing fault classification; ΔE is the increase in evaluation cost (accuracy loss); the temperature T gradually decreases during the search process according to the factor η.

After completing the compression process, the model can be retrained with the new weight structure to fine-tune the weight and obtain better accuracy. Optimization techniques are crucial for efficiently training deep neural networks (DNNs), particularly CNNs. Gradient centralization (GC) [36] is a simple and effective optimization technique for the DNNs that operates directly on gradients by centralizing the gradient vectors to yield a zero mean. It can accelerate the training process and improve the final generalization performance of DNNs. GC is simple to implement and can be easily embedded into existing gradient-based DNN optimizers with only a few lines of code. Furthermore, it can be used directly to fine-tune pre-trained DNNs. GC can be regarded as a projected gradient descent method with a constrained loss function. The Lipschitzness of the constrained loss function and its gradient is better, affording a more efficient and stable training process [37]. GC has been demonstrated to consistently improve the performance of DNN learning, particularly in image classification tasks, which is of the same characteristics as the task of bearing fault diagnosis in this study.

## 3. Experimental Implementation

The experimental testbed is satisfied with the requirements serving for bearing diagnosis experiments and widely used in previous studies [38,39,40]. Briefly, the system is assembled by one drive-end and one non-drive-end shaft. The driving force is transmitted via the shafts by the connection of a gearbox. Each of those shafts is fastened onto the foundation by 2 bearing FAG NJ206-E-TVP2. The non-drive-end shaft is connected to an adjustable blade, acting like a load of the system via a pulley, as shown in Figure 4.

Seven bearing fault types were created: outer race defect (ORD), inner race defect (IRD), roller defect (RD), inner and outer race defect (IORD), outer race and roller defect (ORRD), inner race and roller (IRRD), and inner race, outer race, and roller (IORRD), as depicted in Figure 5.

The data acquisition system consists of a wideband AE sensor attached to a non-drive-end bearing. AE signals from the sensor are acquired by PCI-2-based system at a sampling rate of 150 kHz. The AE sensor (PAC WSα) has an operating frequency range from 100 to 900 kHz, ±1.5 dB directionality, −62 dB peak sensitivity, and the resonant frequency is 650 kHz. Moreover, the PCI-2 board is able to acquire signals at the highest sampling rate of 10 MHz when using one channel; its dynamic range is more prominent than 85 dB. The ADC resolution is 18 bits with a sampling rate of up to 40 MHz. The acquired AE signals from one no-defect (ND) state and seven defective bearings in case of the bearing have the smallest crack size (length = 3 mm, width = 0.6 mm, depth = 0.3 mm), were used to create the dataset used in our experiments. Previous related research shows that small crack sizes on bearings make it challenging to detect faults [38,39,40].

The dataset was categorized into three subsets: the training, validation, and testing subsets. The training and validation subsets were used for the training; the testing subset was used for evaluating our model performance. The samples in the training subset are independent of the rotational speed to the validation and testing subsets, as shown in Table 1. The total number of samples used for the training process (80% for training subset and 20% for validation subset) was calculated by $N_{Classes}\times N_{Samples}\times N_{Speed}$ = 4800 samples, where $N_{Speed}$ = 6, $N_{Classes}$ = 8, $N_{Samples}$ = 100 ($N_{Samples}$ is the total number of signal samples at a specific bearing condition and specific shaft speed).

## 4. Experimental Results and Discussion

Initially, each signal in the training subset was decomposed by the EEMD. Nine IMFs were obtained for each signal segment after stopping because the energy ratio was higher than the max–energy-ratio criterion. The few first IMFs contain the most meaningful information, so their correlation coefficient to the original signals may be higher compared with the remaining ones. Table 2 shows the correlation coefficients between the first nine IMFs and the original signal. We observed that the top-3 IMFs had a strong correlation with the original signal compared with the others. Therefore, the top-3 IMF orders were RIMFs, which were then used as individual signals to evaluate their extent in conveying the bearing fault features.

For the first RIMF comprising 30 signal samples, each type of single fault was used for the training and testing of Efficient-Net to obtain the average classification accuracy (ACC) score. Similarly, the second and third RIMFs were evaluated. Table 3 shows that the first two IMFs achieved high ACC scores. This means that they comprised useful features for classifying bearing faults. Therefore, the first two IMF orders (IMF1 and IMF2) were used to create spectrogram images in the next steps.

### 4.1. Ability of the Proposed Method in Reducing the Size of Representation Image

In common, to evaluate the performance of a diagnosis method, sensitivity in the classification task seems like one of the most important indicators. For this reason, we can see the comparisons based on sensitivity in previous related studies. Hence, we used the sensitivity value as the primary index to evaluate and compare our method with previous methods. Sensitivity is calculated using Equation (2):(2)$$Sensitivity=\frac{N_{True\_Positive}}{N_{True\_Positive}+N_{False\_Negative}}\times 100(\%)$$
where $N_{True\_Positive}$ is the number of correctly classified samples from a particular class, and $N_{False\_Negative}$ is the number of incorrectly classified samples from a particular class.

The average classification accuracy (ACA) was measured using Equation (3) to determine the average classification performance of each dataset:(3)$$ACA=\frac{\sum \mathrm{Sensitivity}}{\sum N_{Classes}}$$
where ∑Sensitivity is the sum of class-wise accuracy for a specific dataset.

In the first experiment, we set the scaling of MobileNet-v2 at 0.1 (called 0.1 × MobiletNet-v2) without using any pruning technique to analyze the relationship between the accuracy of the model and the required system resources when the size of input images changes. The results in Table 4 show that the proposed method achieved high prediction accuracy in classifying eight classes of bearing states (seven types of single and compound faults and one normal state) under variable rotational speeds. Furthermore, the results indicate that using larger image sizes can yield higher prediction accuracies. This is because the model trained on smaller images will learn fewer features than one trained on larger images. Meanwhile, the model using smaller images can be trained and tested for each sample faster. In other words, it uses fewer computational resources. Therefore, the essential features in the representation images can be maintained via a prior decomposition. This approach utilizes the advantage of signal processing algorithms in reducing computational resources. We observed that when the input images were resized from 96 × 96 to 64 × 64 or 32 × 32, the number of the MACs decreased significantly from 6,005,248 to 2,674,688 and 676,352, respectively. Meanwhile, the prediction accuracy decreased slightly and reached a high value between 99.58% and 99.79%. The number of training epochs increased from approximately 50 to 100 when the training model with an image size of 96 × 96 was changed to one with 32 × 32. This means that the small-input model required more training time, but it did not affect the testing process in the online phase.

### 4.2. Compression Ability Via Proposed Scaling and Runing Model

We conducted an experiment to consider the effects of changing the scaling rate and pruning rate (sparsity rate) of the model to analyze the changes in accuracy and required resources. The sparsity rate is the parameter to determine the degree of auto pruning. Throughout our experiments, the parameters for auto compress pruner are set as depicted in Table 5.

First, we used 0.2 × MobileNet-v2. The results in Figure 6a show the efficiency of the model, both in terms of accuracy and system resources. A sparsity rate of 0 means that the original 0.2 × MobileNet-v2 was used. In that case, the average accuracy was 99.79%. Next, we consecutively pruned 0.2 × MobileNet-v2 with sparsity rates from 0.1 to 0.3. We observed that when selecting an appropriate sparsity rate, the prediction accuracy of the CNN model was improved as well because the meaningless constraints between neurons in the network were removed. Specifically, for the case where the sparsity rate was 0.1, the accuracy of bearing fault diagnosis reached 100%. After the accuracy reached the peak of the prediction accuracy, if we continue to increase the sparsity rate, both the system resource and accuracy will decline gradually in a tradeoff manner. Next, we conducted a similar experiment using 0.1 × MobileNet-v2. The results are shown in Figure 6b illustrate the same trend as that of the previous model scale. It is noteworthy that both the number of MACs and the number of parameters decreased significantly compared with the model scale of 0.2 while maintaining high accuracy. With a sparsity rate of 0.1, the accuracy reached a peak of 99.79%, the number of MACs decreased to 663,800, and the number of parameters was reduced to 84,208.

We continued to reduce the scale of MobileNet-v2 until the lowest system resource was determined while maintaining the prediction accuracy. Finally, the best scale was determined to be 0.01. Next, pruning was applied in the same manner as in the two previous experiments. Table 6 shows that the proposed method is better than the previous modern methods not only in terms of prediction accuracy but also in reducing computational resources and memory requirements. After pruning at the best sparsity rate, the model maintained the accuracy of 99.58%, whereas the required resources were less compared with when using LeNet5-based architectures, which are simple CNN models. Two simple CNN-based fault diagnosis methods of Tra et al. [20,21] are used in our comparison. Those methods use the representation of AE signals in the frequency domain to stack each segment on top of each other (two-dimensional energy distribution maps (EDMs)). The author utilized CNN architectures based on Lenet-5 for the feature extraction task. After this, the method [21] uses a hybrid ensemble MLP–SVM classifier to classify bearing faults from extracted features. Meanwhile, in the method proposed in [20], classification was performed by using a common MLP. In addition, a stochastic diagonal Levenberg–Marquardt algorithm was used to enhance the training process. The confusion matrix in Figure 7 illustrates the bearing fault diagnosis for each bearing state class in the case of using 0.01 × MobileNet-v2. As shown, all of the classes yielded highly accurate predictions.

In addition, we proved that the proposed method is suitable for an embedded system by measuring the actual inference time of the classification task when implementing the algorithm on a specific embedded system named Raspberry Pi 3 (Quad-Core 1.2 GHz Broadcom BCM2837 64-bit CPU, 1 GB RAM) [41]. The results in Table 7 show that it is feasible to implement our proposed method on a generic embedded system operating in a short inference time. In the case of 0.01 × MobileNet-v2, the inference time was shorter than 100 ms.

### 4.3. Consistency Ability in Noisy Environments

The stability of a bearing fault diagnosis method is crucial because of the presence of noise in real operating environments. We conducted an experiment to evaluate the consistency ability of the proposed method at a scaling rate of 0.01 in noisy environments. First, the original bearing signal was added to white noise in different ratios from 10% to 30% of the signal amplitude. The average accuracy decreased according to the noise level. Although the results in Table 8 show that the proposed method is more sensitive to noise compared with the previous methods using large input image sizes [38], the accuracy remained high, i.e., 95.63–96.25%. EEMD contributed positively to the acquired IMFs after decomposing in reducing white noise compared with the original EMD. Therefore, we can conclude that the proposed method is reliable at low noise levels.

## 5. Conclusions

We proposed a practical approach to design a CNN-based bearing fault diagnosis process for embedded systems. The efficiency yielded by our proposed method was attributed to the advantages of signal processing methods and the absence of the disadvantages of modern CNN architectures when it was applied to limited-resource systems. The advantages of signal processing methods were illustrated when representing nonstationary signals containing the bearing status at variable rotational speeds. In addition, EEMD and CNN-based IMF selection were suitable for simplifying signals and extracting useful components even in low noise environments, contributing to the decreased image input size of the CNN model. The pruning approach applied for the state-of-the-art MobileNet-v2 at various scaling rates proved that reducing system resources based on knowledge regarding the trained model is effective for bearing fault diagnosis. The results of our experiments indicated that using the proposed method, the maximum attainable accuracy in compound bearing fault diagnosis was 99.58%, with the consumed system resources less than those of the LeNet5-based method. The implementation of the CNN model on the Raspberry Pi 3 embedded system demonstrated the feasibility and reliability of the proposed method for use in industrial embedded systems for compound fault diagnosis under variable rotational speeds.

## Figures and Tables

**Figure 1 sensors-20-06886-f001:**
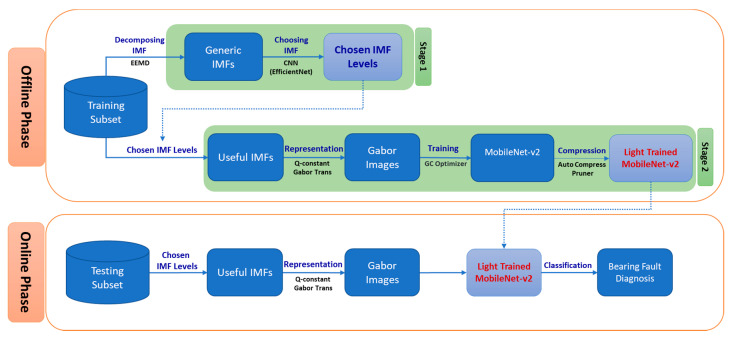
Proposed bearing fault diagnosis method for embedded systems.

**Figure 2 sensors-20-06886-f002:**
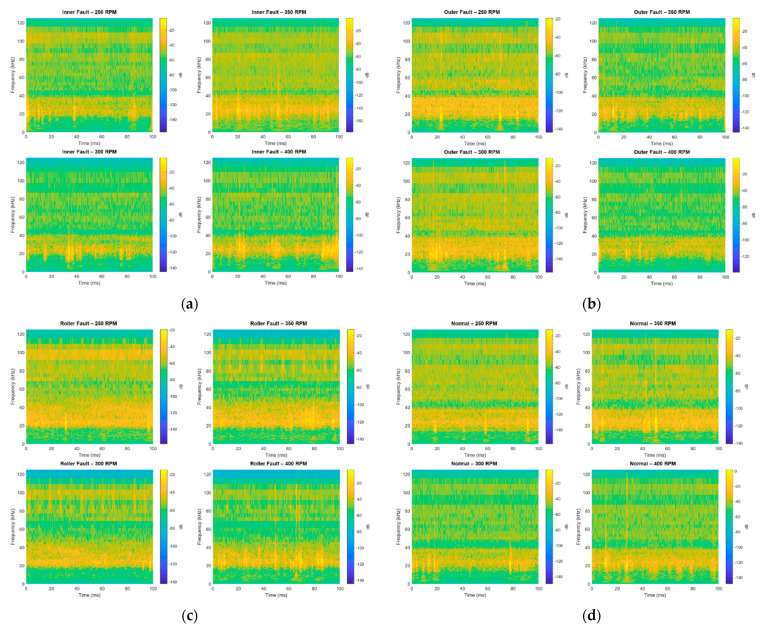
Spectrograms created by constant-Q transform for the acoustic emission (AE) signals from the bearings at various states: (**a**) inner race defect; (**b**) outer race defect; (**c**) roller defect; (**d**) normal state.

**Figure 3 sensors-20-06886-f003:**
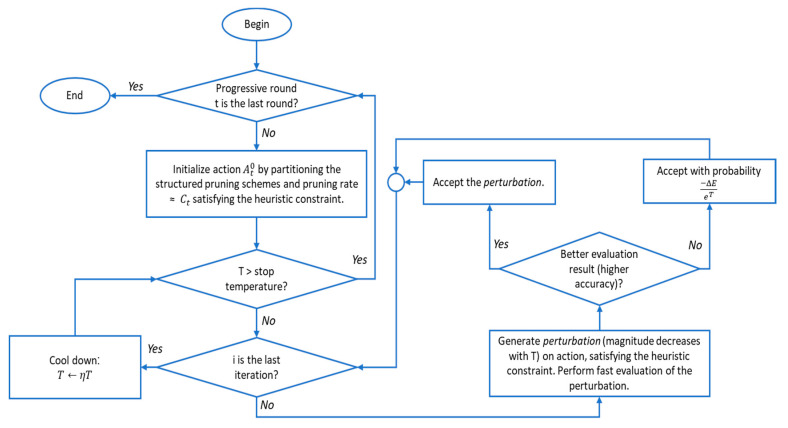
Flowchart of auto-compression algorithm.

**Figure 4 sensors-20-06886-f004:**
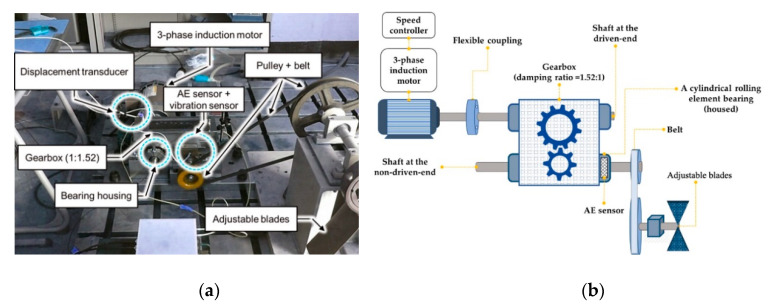
Testbed for collecting AE signals from compound faults: (**a**) real testbed component positions; (**b**) testbed diagram.

**Figure 5 sensors-20-06886-f005:**
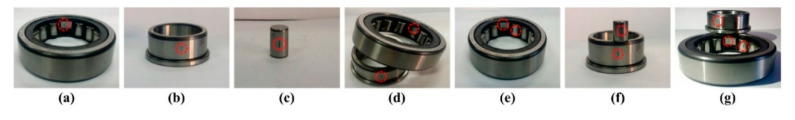
Single and compound bearing faults: (**a**) outer race defect (ORD); (**b**) inner race defect (IRD); (**c**) roller defect (RD); (**d**) inner and outer race defect (IORD); (**e**) outer race and roller defect (ORRD); (**f**) inner race and roller (IRRD); (**g**) inner race, outer race, and roller (IORRD).

**Figure 6 sensors-20-06886-f006:**
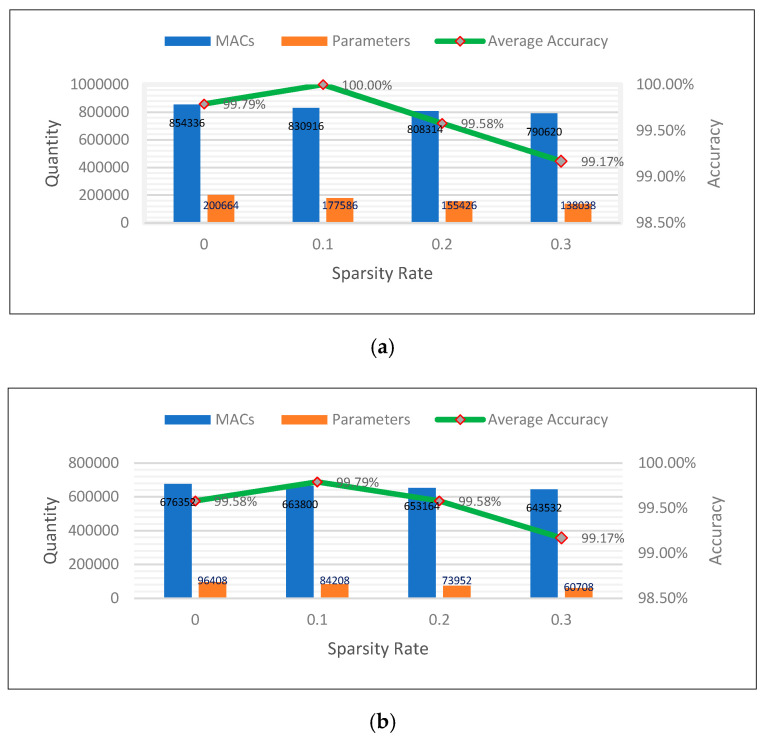
Results of after training and pruning model at different sparsity rates and scaling rates:(**a**) 0.2 × MobileNet-v2 (scaling rate = 0.2), (**b**) 0.1 × MobileNet-v2 (scaling rate = 0.1).

**Figure 7 sensors-20-06886-f007:**
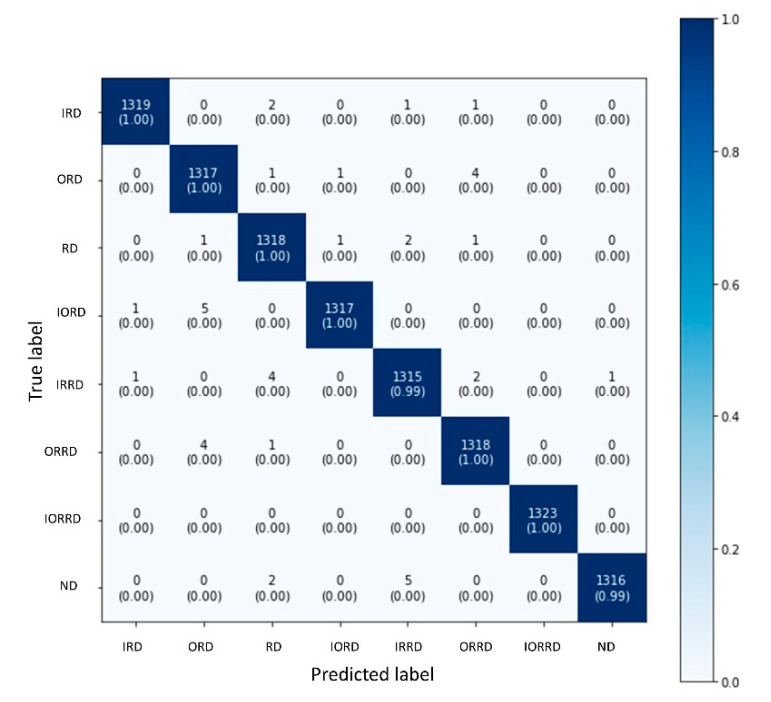
Confusion matrices for testing results of the proposed method at a scaling rate of 0.01.

**Table 1 sensors-20-06886-t001:** Dataset for bearing fault diagnosis.

Single and Compound Bearing Failures		Rotational Speed (rpm)	Crack Size		
			Length (mm)	Width (mm)	Depth (mm)
Dataset	Training subset	300, 400, 500	3	0.60	0.30
	Validation subset, Testing subset	250, 350, 450			

**Table 2 sensors-20-06886-t002:** Correlation coefficients between the first nine intrinsic mode functions (IMFs) and the original signal.

	IMF1	IMF2	IMF3	IMF4	IMF5	IMF6	IMF7	IMF8	IMF9
CC score	0.8655	0.7048	0.0865	0.0002	0.00008	0.0001	0.00003	0.0001	0.0004

**Table 3 sensors-20-06886-t003:** Average classification accuracy (ACC) scores of related IMFs (RIMFs).

	IMF1	IMF2	IMF3
ACC score	0.942	0.933	0.253

**Table 4 sensors-20-06886-t004:** Results of training proposed model with various image sizes.

Input Image Size (pixel)	MACs	Parameters	Average Accuracy	No. of Epochs
96 × 96	6,005,248	96,408	100.00%	50
64 × 64	2,674,688	96,408	99.79%	70
32 × 32	676,352	96,408	99.58%	100

**Table 5 sensors-20-06886-t005:** Configurations for auto compress pruner.

ADMM-Based Structured Pruning	Initial penalty parameter *ρ* = 1 × 10^−4^
	Training epochs for the optimization of ADMMPruner: 5
	Number of iterations of ADMM Pruner: 5
	Optimizer: gradient centralization (GC)
Simulated Annealing (SA)	Cooling factor: *η* = 0.9
	Initial perturbation magnitude to the sparsities: 0.35
	Start temperature of the simulated annealing process: 100
	Stop temperature of the simulated annealing process: 20

**Table 6 sensors-20-06886-t006:** Diagnosis performance of the proposed method at a scaling rate of 0.01 and consumed system resources compared with existing methods.

0.01 × MobileNet-v2	MACs	Parameters	Average Accuracy
Original	602,096	46,056	99.58%
Sparsity rate: 0.1	596,816	41,304	99.58%
Sparsity rate: 0.2	592,776	37,668	99.38%
Sparsity rate: 0.3	589,174	34,788	98.95%
LeNet5-based methods	MACs (LeNet5)	Parameters (LeNet5)	Average Accuracy
Method [21]	668,272	72,376	98.74%
Method [20]	668,272	72,376	94.20%

**Table 7 sensors-20-06886-t007:** Inference time on Raspberry Pi 3 of convolutional neural networks (CNN) at various scaling rates.

Models	Inference Time Per Sample on Raspberry Pi 3 (ms)
0.2 × MobileNet-v2 (original)	120
0.1 × MobileNet-v2 (original)	100
0.01 × MobileNet-v2 (original)	90

**Table 8 sensors-20-06886-t008:** Results of compound bearing fault diagnosis in various noise ratios.

Noise Levels	ACC
10%	96.25%
20%	96.04%
30%	95.63%

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
