# Peer review of "Deep Learning-Based Bearing Fault Diagnosis Method for Embedded Systems"

_sensors, 2020, doi:10.3390/s20236886_

Round 1

Reviewer 1 Report

The early fault detection of rolling-element bearings is essential in machine health monitoring. This paper proposes a novel approach for establishing a CNN-based process for bearing fault diagnosis on embedded devices using acoustic emission signals. The experimental results show that the proposed method can reduces the computation costs while maintain the high classification accuracy. The comments are listed as follows.
(1)How to determine the scaling of MobileNet-v2? It is suggested that the scaling value or scaling rate 0.1 or 0.2 should be consistent with the description of Subsection 2.4.
(2) How to determine the size of the input images?
(3) What is the meaning of the sparsity rate? It is suggested that this term should be specified in the text and explain tow to conduct the sparsity rates from 0.1 to 0.3?
(4)There are some English writing errors such Those method in Line 425 of Page 12 should be "Those methods". And, A stochastic should be And, a stochastic.

Author Response

Please see the attached PDF file containing the summary of changes according to your valuable comments.

Reviewer 2 Report

The paper is well written and structured.

The sections and the goals are clear and detailed. I agree with the training procedures and dataset split procedures.

I can see that many time and resources have been spent in signal pre-processing and this leads to my main concern regarding this piece of work. You clearly stated in each section the importance of maintaining the inference process as lightest as possible.

Nevertheless, to reach your goals, you decided to pick up a CNN as a fault predictor. It is well known how heavy are these architectures from the memory storage point of view, so why did you choose it to perform your experiments? You could pick up another DNN model (as it has been already done in a closely related work [1]) and you would have counted on lower memory occupancy and the fast inference time, crucial factors for an embedded application. It is highly recommended to investigate this opportunity.

Moreover, I cannot figure out the advantage of coming from the 2D image conversion of the pre-processed signal. Why you did not divide your dataset into healthy and faulty condition sets and then apply a DNN model directly on these data?

I invite the authors to enrich the literature review including [1] and address my feedback for improving the quality of the manuscript. [1] Cipollini, F. and Oneto, L. and Coraddu, A. and Savio. S., Data-Enabled Discovery and Applications, Num:1 - Unsupervised Deep Learning for Induction Motors Bearings Monitoring, Vol:3 - 2018.

Author Response

(The authors gave the same response as above.)

Reviewer 3 Report

This paper does a nice job of explaining an embedded system for the evaluation of ball bearing manufacturing problems via analysis of vibrational and related information in the testing process.  The methodology uses convolution neural nets, a technology that is maturing very rapidly and has lots of examples of workable applications that are related to these types of problems. The study is well laid out and explained, and the qualitative aspects of the study are well put together, affirming a high accuracy rate of fault detection.

However, at present the presentation in this paper is definitely inadequate. Though the English is checked for grammar and for word usage at a local level, the intermediate level understandability of sentences and use of the English clearly indicates that the authors are not experienced in writing in English. It is clear that this paper, if it is to be rewritten, will need to be edited by subject experts who are familiar with the material as well as English; it is not sufficient for editors who do not understand its lack of comprehensibility as far as technical content. More specifically, there are editing services that fulfill the functions I am describing here, but these involve writers who have some experience in writing scientific material in English. As the paper stands right now, it is certainly not publishable.

Author Response

(The authors gave the same response as above.)
